# Surveillance of a Pest Through a Public Health Information System: The Case of the Blackfly (*Simulium erythrocephalum*) in Zaragoza (Spain) during 2009–2015

**DOI:** 10.3390/ijerph17103734

**Published:** 2020-05-25

**Authors:** Ignacio Ruiz-Arrondo, José A. Oteo, Javier Lucientes, Ana Muniesa, Ignacio de Blas

**Affiliations:** 1Center for Rickettsiosis and Arthropod-Borne Diseases, Hospital San Pedro-CIBIR, Piqueras Street 98, 3rd floor, 26006 Logroño (La Rioja), Spain; jaoteo@riojasalud.es; 2Faculty of Veterinary Sciences, Instituto Agroalimentario de Aragón (IA2), Universidad de Zaragoza, Miguel Servet Street 177, 50013 Zaragoza (Aragon), Spain; jlucien@unizar.es (J.L.); animuni@unizar.es (A.M.); deblas@unizar.es (I.d.B.)

**Keywords:** insect bites, OMI-AP, outbreak, primary care, simuliidae, *Simulium erythrocephalum*

## Abstract

*Background*: Animals and people in many Spanish regions are increasingly being affected by blackfly bites in the last decade. Because of blackflies, the city of Zaragoza has become in recent years a paradigm of discomfort in Europe, with thousands of citizens affected. The OMI-AP system (Stacks, Barcelona, Spain) implemented by the Government of Aragón, a software that manages the electronic medical history of all patients, has been evaluated in order to document the increase of insect bite recorded by the primary care consultations in Zaragoza after the first outbreak of blackflies occurred in 2011. *Methods*: An observational, ecological and longitudinal study of insect bites recorded at the primary care consultations was carried out in primary care during the period 2009–2015. *Results*: The incidence of medical consultations by insect bites in Basic Health Areas (BHA) near to rivers is higher than the furthest BHA. Rural BHA are more affected by insect bites than the urban ones. The increase of medical assistance due to insect bites in Zaragoza since 2011 is correlated with the blackflies bites. *Conclusions*: This tool was very useful to describe the initial stage of this public health problem. It could be used for guiding public health responses in terms of surveillance and management of this pest.

## 1. Introduction

The public health importance of blackflies in Europe is primarily attributed to nuisance and to the biting activity of females [1]. The changes through human activities like agricultural land use and the improvement of water quality, among other understudied reasons that affect the river ecosystems, have enabled an increase of blackfly populations in the middle and lower stretches of several Spanish rivers, reaching a pest status [2]. Therefore, in some regions in Spain, blackfly outbreaks have an increasing frequency, threatening the public and animal health [2,3,4].

Of the 54 reported and two in doubt blackfly species in Spain [5], only some of them have been identified as pest species, probably due to the lack of integrated research approaches in the Spanish blackfly taxonomy [6]. *Simulium erythrocephalum* is currently the most important anthropophilic pest species in Spain [7] and in some other European countries like Serbia [8]. The city of Zaragoza (Aragón, Spain), with less than 700,000 inhabitants, has become in the last decade a paradigm of annoyance caused by blackflies in Europe [7]. The bites of *S. erythrocephalum*, the species responsible for blackfly outbreaks in the city, sometimes causes allergic reactions that require medical assistance. Since 2011, problems due to arthropod bites have increased more than 200% at the region’s health centers [7].

Public health administrations implement information systems to identify health problems of the human population and to generate reliable information from health and non-health data sources that allow them to design specific health programs. One of these information systems is the one that collects the electronic medical records of the patients. In Aragón, it is collected and managed through a software called OMI-AP, where health conditions are coded according to the International Classification of Primary Care Diseases (CIAP-1). In this way, the OMI-AP system becomes an extremely useful tool to record data from the notifiable diseases and other processes under surveillance [9].

In Spain there is no active surveillance regarding arthropod bites to assess the public health impact in case of any observed changes in arthropod activity. The Mosquito Alert app is a useful citizen science tool to study the spread and establishment of the invasive mosquito *Aedes albopictus* in Spain [10], but it records the presence of the vector and not insect bites. We have some examples of increase of *Ae. albopictus* bites requiring healthcare in the Spanish Mediterranean coast [11,12] and a slight increase in the Basque Country [13]. However, to our knowledge, despite the increasing public health problem attributable to the blackfly pest, there are only a few published reports about the incidence of blackfly bites requiring healthcare attention in some Spanish regions. Two examples are the previously mentioned problems suffered by the city of Zaragoza [7] and a study developed in Catalonia, where 1,484 medical attentions to blackfly bites were recorded from the end of May to the end of October 2005, the first year in which massive bites by this insect were detected [14,15].

There are few examples of either a national surveillance system or published studies investigating the epidemiology of arthropod bites in Europe [16]; and as far as we know, there is not one about blackfly bites. It is of paramount importance to study the epidemiology of pest bites that involves massive bites to the population, so as to know the health determinants involved; this could represent a valuable step in order to plan new interventions and subsequently evaluate the impact of the control measures established. For this reason, the aim of this work was to use the OMI-AP health information system implemented by the regional government to assess the increase of primary care electronic arthropod bites records and to describe the dynamics of blackfly pest through its impact in the primary care in Zaragoza during the period 2009–2015.

## 2. Materials and Methods

An observational, ecological and longitudinal study of insect bites in primary care consultations was carried out during the period 2009–2015 in Zaragoza city.

### 2.1. Study Area

The study was carried out in Zaragoza and its metropolitan area, which with 664,938 inhabitants is the capital of Aragón, a region that is located in the northeast of Spain. Three rivers—Ebro, Gállego and Huerva—run through the city. The Ebro is the second longest river in Spain (930 km), with the highest average flow (600 m^3^/s, but of an irregular nature). At the end of summer, it presents very low water levels, typically carrying less than 10% of its average flow. Most of the blackfly breeding sites were identified along the Ebro River downstream and upstream from the urban area of Zaragoza [2,7]. The other two rivers have less flow than the Ebro. The Huerva River runs underground as it passes through the city, while along the Gállego River there is a paper mill industry upstream from the urban area that emits cellulose pulp which covers all the river substrates, preventing larval attachment.

No identifications of blackfly species were made in this study. The species responsible for human bites in Zaragoza is inferred to be *S. erythrocephalum* based on previous studies carried out during our same study period [7,17]. Species of the subgenus *Wilhelmia* have been also detected in the outskirts of Zaragoza city, but they have not been identified in human biting cases [7] so their role as a biting pest of humans is not discussed.

### 2.2. Data Source

The Public Health department of the Government of Aragón (through a model of commitment to adapt to the Organic Law on Protection of Personal Data (LODP in Spanish), provided us the data from the OMI-AP system on primary care consultations for any reason and on insect bites consultations in primary care. All data were disaggregated by BHA belonging to the metropolitan area of Zaragoza city in the period 2009–2015. The total number of primary care consultations and the insect bites primary care consultations were obtained on a monthly and weekly basis respectively.

The BHA are the territorial assistance units where the different primary care teams are grouped. Each BHA has one or more primary care centers. We selected a total of 41 BHA. Of them, 33 BHA are part of Zaragoza and categorized as urban BHA (in blue, Figure 1); we also selected the villages within a radius of 30 km from the capital and these were categorized as rural BHA (in yellow, Figure 1). Appendix A lists each BHA and their corresponding zone (urban or rural). It must be taken into account that people may have been bitten in any area of the city but that patients normally go to the BHA that corresponds to their place of residence and not where they have been bitten.

Arthropod bites requiring healthcare were obtained from the primary electronic medical record coded as S12 “insect bite” according to the International Classification of Primary Care (ICPC-1). Data were debugged deleting those descriptors that corresponded to specific health intervention where the causative arthropod was detailed and not related to the objective of our study (e.g., tick, wasp, etc.). In total, for each arthropod bite record requiring healthcare we used the following variables: year, month, week, BHA and zone type (rural or urban). Data were imported in a database managed with Access 2016 (Microsoft, Readmon, WA, USA) and then SQL queries in order to recode, categorize and create new variables.

### 2.3. Data Analysis

Firstly, insect bites in primary care consultations/10,000 primary care consultations for any reason were calculated. They were grouped with different criteria, temporal (year, month, week) and spatial (BHA and zone). The numbers of insect bite in primary care consultations were analyzed annually and monthly, considering the spatial distribution according to rural/urban zone.

The mean, standard deviation(s) and range of values (minimum and maximum) were used to describe the quantitative variables, evaluating the normality of the data with the Kolmogorov-Smirnov test. To assess the spatio-temporal variation of the bite rates we used ANOVA test to assess differences among years, and later we used Duncan post hoc test to evaluate differences between years. In order to compare differences between urban and rural zones, we used the Student’s test for independent samples. In all cases a weighted analysis was carried out using the number of primary care as weighting variable.

Analysis was performed using SPSS 19.0 for Windows (IBM, Armonk, NY, USA) setting the alpha error to 0.05. Graphs were generated with Microsoft Excel 2013 (Microsoft, Readmon, WA, USA). Figure 1, Figure 2 and Figure 3 were created by QGIS V.3.0.1 (QGIS Development Team 2018, Available from: http://www.qgis.org). Ranges used in the Figure 2 and Figure 3 were chosen using the minimum and maximum values in the period 2009–2015 and subsequently calculating the ranges based on the quartiles of the year 2012 as it reached the maximum value of the entire series.

## 3. Results

### 3.1. Analysis of Insect Bites Primary Care Consultations Accumulated Annually

The incidence of insect bites in primary care is given in Table 1. Over the years, there was an increasing evolution in the number of insect bites in primary care consultations/10,000 primary care consultations compared to 2009, reaching a maximum in 2012. Later, from 2013 to 2015, a decrease in the number of insect bites medical records is observed. The high value of the mean in 2009 is explained by the fact that the registration of total primary care consultations is less than in subsequent years because the electronic system for registering medical records was still being implemented (Table 1). An increase of 217% was observed in the number of insect bites in primary care consultations/10,000 primary care consultations in 2011 compared to the previous years 2009 and 2010. This increase was even greater during 2012, reaching 280%. Subsequently, from 2013 to 2015, a decrease in the incidence of insect bites records was observed with respect to the years 2011 and 2012, but remaining above the figures in the years 2009 and 2010. Regarding the zone type, significant differences were observed between rural and urban zones, being greater the number of consultations in the first zone (Table 1).

The spatial analysis of insect bites incidence confirms what was previously observed in the statistical analysis (Figure 2 and Figure 3). In 2009, several BHA are on grey (outside the established range green-red scale) and yellow (Figure 3) because the electronic system for registering medical records was not totally implemented in all BHA (Table 1). There is also a BHA in red during 2010 for this same reason (Figure 2). Figure 3 shows the highest incidence of insect bites in BHA close to the Ebro River during the years 2011 and 2012. In addition, an increase in the number of insect bites consultations already occurred in 2009 and 2010 in rural zones downstream of Zaragoza (Figure 2). While urban BHA have the lowest number of insect bites records in the 2013–2015 period (Figure 3), several rural BHA remain at high levels (Figure 2). Rural BHA (Zuera and María de Huerva, ID: 22 and 18 respectively in Figure 1), distant from the Ebro River but close to where the Gállego and Huerva rivers’ flow, respectively, had a large number of insect bites consultations in addition to a different annual pattern of consultations (Figure 2).

### 3.2. Analysis of Insect Bite Primary Care Consultations Accumulated Monthly

Table 2 shows how the highest insect bites in primary care consultations occurred during the period July–September in 2009, July and August in 2010, and peaked in August and July respectively. However, during the following years (2011 and 2012), the highest incidence of insect bites records began in May, lasting until August. In both years, the maximum was reached during the month of June, exceeding the threshold of 150 and 250 insect bites in primary care consultations/10,000 primary care consultations in 2011 and 2012 respectively. Subsequently, in 2013 a similar pattern to that of 2010 was observed, with the beginning and maximum of consultations in July. In the remaining years 2014 and 2015, the monthly pattern of consultations was modified again, reaching the peak of insect bites in primary care consultations/10,000 primary care consultations in June, as the years 2011 and 2012.

There were differences between the monthly insect bites in primary care consultations in rural and urban zones (Figure 4). Overall, during the study period, it was observed that the rural zone had higher insect bites incidence than the urban; except for 2012, when both zones showed a similar pattern.

## 4. Discussion

The increase in the number of insect bites in primary care consultations in the metropolitan area of Zaragoza during the period 2011–2015 compared to 2009–2010 is due to the massive presence of blackflies on the banks of the Ebro River as it flows through Aragón [2,17]. The fact that BHA next to the Ebro River were the most affected reaffirms the hypothesis that the anthropophilic blackflies have their main breeding sites in this river. In addition, the massive presence of this pest is also responsible for the change in the monthly pattern of insect bites medical records observed from 2011 with respect to the period 2009–2010. It could be thought that the monthly fluctuation of consultations in the period 2013–2015 (Table 2) is due to the blackfly control management, established in the Ebro River as it flows through the urban area of Zaragoza from 2011, and the habituation of citizens to insect bites. However, in 2017, another significant blackfly outbreak occurred in Zaragoza, exceeding the number of insect bites in primary care consultations since 2012. In the first six months of 2017, 11,512 insect bites in primary care consultations were recorded in Aragón, a higher number than the 9,422 consultations for the same period in 2012 [18], the year with the highest number of insect bites in our series. Therefore, we think this temporary pattern in the insect bites records is due to changes in the blackfly population density that in turn depends directly on the increased population along the rivers.

During the first outbreak in 2011, Ruiz-Arrondo et al. [17] observed the immature stages of *S. erythrocephalum* using macrophytes as a substrate to develop their biological cycle. Submerged macrophytes are phanerogamic plants that live rooted or not on the river bed and maintain their entire structure underwater, being able to grow to occupy large extensions of the river bed [19]. This association between the massive proliferation of macrophytes and blackflies has also been observed in other Spanish regions affected by this pest [2,14,20,21,22]. This relationship of macrophyte-blackfly larvae was observed again in 2012 in the Ebro River in the vicinity of Zaragoza [23]. From 2013 to 2015, the incidence of insect bites was much lower because the hydrology of the Ebro River changed compared to 2011–2012. Overall, a lower presence of macrophytes was detected than in 2011–2012 and therefore, the blackfly population was lower than in previous years (personal observation, Ruiz-Arrondo). The hypothesis proposed is that prolonged high flows of the Ebro River during the winter and the beginning of spring negatively affect the breeding sites of blackflies, which entails a decrease in the population of blood-sucking females and consequently in the number of insect bites attended at primary care centres. Marqués [21] also espoused that the flows below the ecological threshold in the Ter River in Catalonia support the presence of preimaginal forms of blackflies. The opposite situation has been described in Serbia, where a high flow of the Danube River during spring boosts the occurrence of blackfly outbreaks during the subsequent spring and summer [8,24]. All these observations reinforce the need to carry out a study to know precisely how the hydrology of the Ebro River affects the dynamic population of macrophytes and blackflies in the metropolitan area of Zaragoza.

The increase in consultations in rural zones around the Ebro river, downstream from Zaragoza (Figure 2), observed in 2010 (prior to the first outbreak in Zaragoza city in 2011) indicates that the blackfly pest expansion followed an ascending process from Ebro downstream; a fact also evidenced by Figueras et al. [3] who described the first massive blackfly bites on flocks of sheep in the same BHA zones in 2010, more than 30 km from Zaragoza city. Actually, the first blackfly outbreak recorded in settlements near the Ebro River occurred in 2005 in Catalonia (more than 200 km downstream from Zaragoza) [15]. A plausible explanation for the fact that urban areas have fewer insect bites records than rural zones during the 2013–2015 period is that the main breeding sites for *S. erythrocephalum* in the Ebro River are down- and upstream the urban area of Zaragoza [7]. A significant number of consultations in rural BHA far from the Ebro River but near where the Huerva and Gállego Rivers flow suggests that these latter rivers have blackfly breeding sites and the insect bites medical records in these BHA are strongly influenced by these rivers and not by the Ebro. Larvae and pupae of *S. erythrocephalum* and other species belonging to the subgenus *Wilhelmia* have been identified in both rivers outside the urban area of Zaragoza city [23]. However, the BHA through which the urban section of the Huerva River runs are not as affected by insect bites (Figure 3) because the conditions for the development of the preimaginal forms do not exist [17].

*Simulium erythrocephalum* has been described as a species with great flight capacity. Živković [25] described how this species was able to fly distances of 20-30 km from its breeding sites in search of hosts. Ignjatović-Ćupina et al. [8] explained how females of this species could move to distant localities more than 5 km from their breeding sites along the Danube River. In our case, the flying ability of this species is also confirmed as individuals of *S. erythrocephalum* were caught in human biting acitivty in green areas in the center of Zaragoza 3 km distant in a straight line from the closest point of the Ebro River [23]. This observation shows that its flight capacity is greater since adults have to move through the streets of Zaragoza city on an irregular flight from their breeding sites, located mainly outside the urban area of Zaragoza [7]. The flying ability of *S. erythrocephalum* could also explain why certain BHA away from the axis of the rivers had significant incidences of insect bites. Ignjatović-Ćupina et al. [8] showed how the spatial distribution of adults was irregular in the locations distant from the main blackfly larval foci.

In Europe, there are several countries that have suffered and/or suffer blackfly outbreaks that regularly threaten public health, for example: United Kingdom [26,27], Serbia [8,24,28,29], France [30,31], Hungary [32], Italy [33], Turkey [34] and Scandinavian countries [1], among others. As far as we know, although there are no published European studies based on a temporal analysis of blackfly bite medical consultations, several authors have quantified the number of people who required health care during a blackfly outbreak. Hansford and Ladle [26] described 600 people were attended in healthcare centres due to the bite of the species *S. posticatum* during 4 weeks in the spring of 1972 in the surroundings of Blandford (south UK). In Serbia, *S. erythrocephalum* has produced outbreaks in the province of Vojvodina that required medical services; for example in 1965, during the first recorded outbreak in the country, 37 people needed healthcare [28] and later in 1970, more than 2000 people [24,29]. More recently, in 2006, 30 people required hospitalization [8], although the number of citizens affected was much higher.

Summer is the season when the majority of arthropod groups are more active and consequently there is normally a higher biting activity in primary care during this period. However, blackflies in Zaragoza have caused a change in the insect bite patterns throughout the year, advancing the threshold and changing the peak of insect bite consultations in primary care. Our results indicate that the period with the highest risk of being bitten by insects in Zaragoza ranges from mid-May to the end of August, with June being the month with the maximum frequency of insect bite consultations; coinciding with the highest biting rate of *S. erythrocephalum* recorded during the months of May and June in Zaragoza [7]. Some European studies have pointed out similar patterns. Ignjatović-Ćupina et al. [8] trapped, using CDC traps baited with CO_2_, the most number of females of this species since beginning of April till mid of June in Serbia. In addition to that, the highest insect bite records were registered at the same period [8]. Bardin [31] explained that the nuisances of *S. erythrocephalum* in the south of France started in May decreasing during June and July but continued till October. Other authors have pointed out previously a decreasing in the adult population of this species during the months of July, August and September [8,35,36,37,38].

Traditional human health surveillance systems are unsuitable for monitoring arthropod bite activity as it is not usually an important health problem in terms of prevalence and only a small percentage of cases require hospital treatment [16]. However, in regions where arthropod bites start to threaten public health or it represents an endemic or recurring problem, surveillance systems should be established to monitor its incidence, which in addition to helping in the study of the arthropod bite epidemiology, it would be useful as an evaluation tool of pest management. A step forward would be a real-time syndromic surveillance system to monitor the incidence of arthropod bites as proposed in England by Newit et al. [16].

A limitation of the study is that only a small proportion of blackfly bites require healthcare, and of those that do, not all are attended in the public primary care system. This means that the OMI-AP system is currently detecting only the tip of the iceberg of the significant problem posed by this pest in the surroundings of Zaragoza. In any case, this surveillance system through the analysis of insect bites consultations in primary care is a good tool to assess the impact of this pest on public health and to delimit the period of greatest risk of insect bites.

Further work should explore other public healthcare settings, such as emergency departments in hospitals, analyze the arthropod bite medical records weekly getting deep in other variables such as differences between age, sex and date of attendance, and improve our knowledge of the dynamics of *S. erythrocephalum* such as number of generations per year, among others.

## 5. Conclusions

A systematic analysis of the electronic medical records of the OMI-AP system has shown to be a very useful tool to enable the early identification of a public health problem at the regional level through the epidemiology of arthropod bites attended in primary care. Furthermore, this study can be used to guide public health responses like to assess changes in arthropod biting and to establish the highest risk blackfly biting periods. This study could establish the bases of a syndromic surveillance system to help taking blackfly control decisions, such as establishing a baseline incidence of arthropod bites that marks the beginning of the control season and evaluate the effects of blackfly pest management actions.

## Figures and Tables

**Figure 1 ijerph-17-03734-f001:**
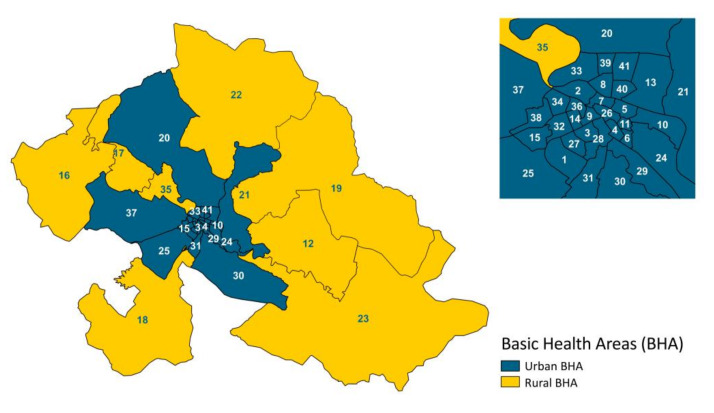
Basic Health Areas (BHA) in the metropolitan area of Zaragoza.

**Figure 2 ijerph-17-03734-f002:**
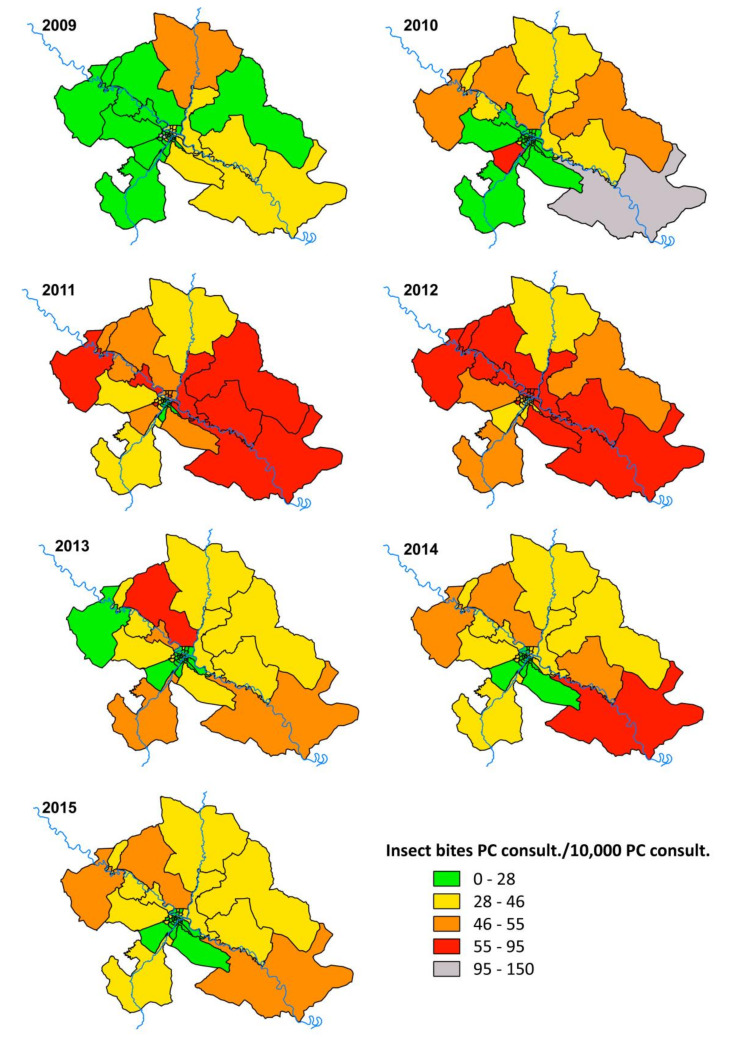
Insect bites primary care consultations/10,000 primary care consultations according BHA in the metropolitan area of Zaragoza during the period 2009–2015. PC: primary care.

**Figure 3 ijerph-17-03734-f003:**
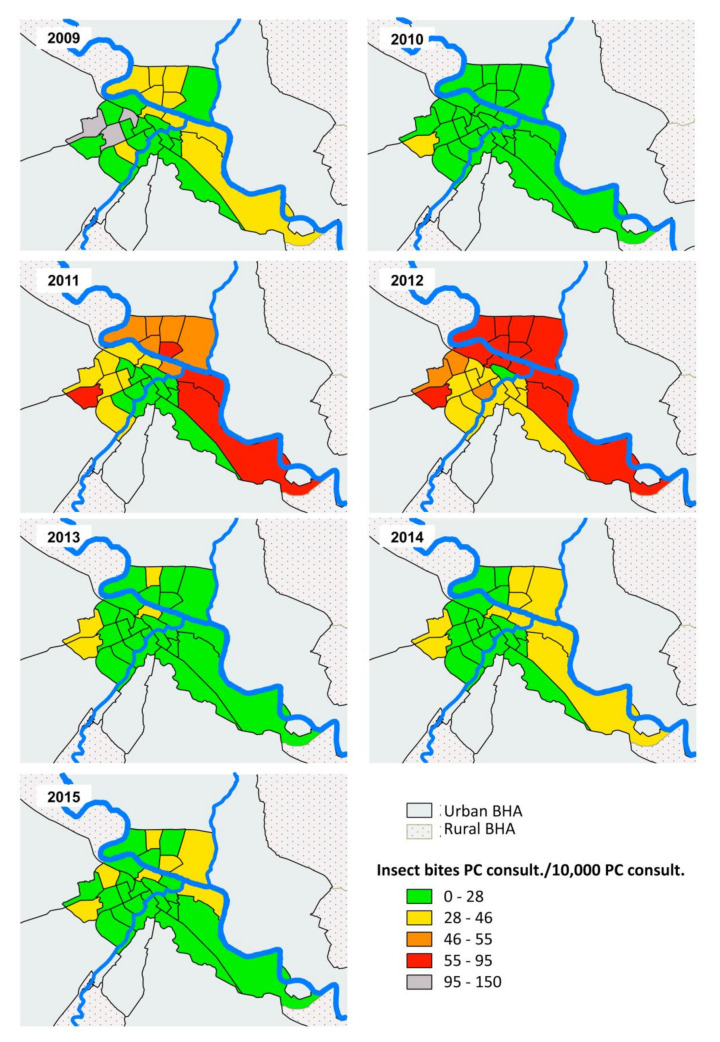
Insect bites primary care consultations/10,000 primary care consultations according urban BHA in the metropolitan area of Zaragoza during the period 2009–2015. PC: primary care.

**Figure 4 ijerph-17-03734-f004:**
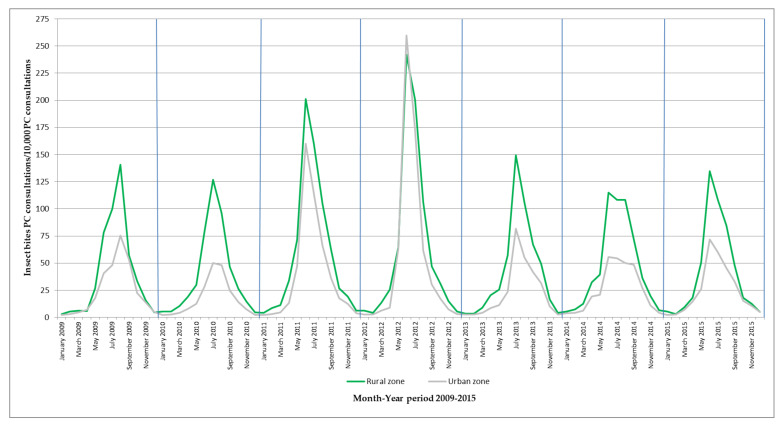
Monthly numbers of insect bites primary care consultations/10,000 primary care consultations by zone in the metropolitan area of Zaragoza during the period 2009–2015. PC: primary care.

**Table 1 ijerph-17-03734-t001:** Insect bites primary care consultations/10,000 primary care consultations in the metropolitan area of Zaragoza stratified by year.

Year	Total PC	Insect Bites PC		Urban		Rural		*p*-Value *
		Mean	s	Mean	s	Mean	s	
2009	1,718,024	20.5 ^b^	28.31	18.9 ^b^	25.46	36.0 ^a^	45.49	<0.001
2010	3,605,585	19.1 ^a^	28.76	16.3 ^a^	20.86	37.9 ^b^	55.18	<0.001
2011	3,766,465	41.5 ^f^	59.51	38.9 ^f^	57.06	57.9 ^f^	70.73	<0.001
2012	3,671,765	53.5 ^g^	85.16	51.9 ^g^	85.36	62.9 ^g^	83.35	<0.001
2013	3,526,776	24.7 ^c^	32.26	21.5 ^c^	27.81	42.1 ^d^	46.41	<0.001
2014	3,522,956	27.6 ^e^	29.17	24.3 ^e^	24.29	45.6 ^e^	43.54	<0.001
2015	3,546,737	26.4 ^d^	31.17	23.7 ^d^	26.84	40.9 ^c^	45.30	<0.001
**Total**	**23,358,308**	**31.5**	**49.69**	**28.8**	**47.37**	**47.4**	**59.36**	**<0.001**
*p*-value **		<0.001		<0.001		<0.001		

* Significance of Student’s *t* test to test differences between zones, ** Significance of ANOVA test to analyze differences between years. Different superindexes indicate significative differences (*p* < 0.05) according the Duncan post hoc test. s is standard deviation. PC: primary care.

**Table 2 ijerph-17-03734-t002:** Monthly pattern of insect bites primary care consultations/10,000 primary care consultations in the metropolitan area of Zaragoza during the period 2009–2015.

Month	2009				2010			2011				2012		
	Mean		s		Mean		s	Mean		s		Mean		s
January	2.2 ^a^		1.76		2.5 ^a^		2.92	2.4 ^a^		1.86		3.2 ^a^		2.57
February	3.2 ^a,b^		2.33		3.1 ^a^		3.05	3.7 ^a^		3.34		3.0 ^a^		2.81
March	4.6 ^a,b^		4.30		5.0 ^a,b^		4.50	5.5 ^a,b^		4.34		7.0 ^a,b^		4.37
April	7.0 ^a,b^		7.46		9.1 ^a,b,c^		11.96	16.2 ^a,b^		12.36		11.4 ^a,b^		9.71
May	18.2 ^c^		12.73		14.7 ^b,c^		11.91	50.5 ^c^		26.81		63.3 ^d^		27.83
June	43.3 ^d^		30.66		34.7 ^d^		36.62	165.7 ^f^		78.27		257.1 ^f^		96.36
July	52.2 ^d^		35.85		60.5 ^e^		52.62	120.7 ^e^		58.75		177.1 ^e^		55.01
August	86.1 ^e^		34.74		54.7 ^e^		33.93	72.3 ^d^		37.46		68.5 ^d^		29.28
September	53.9 ^d^		18.62		27.7 ^d^		16.57	40.0 ^c^		19.99		32.6 ^c^		15.83
October	24.6 ^c^		6.90		16.0 ^c^		10.33	18.8 ^b^		10.38		19.2 ^b^		10.03
November	14.0 ^b,c^		7.08		8.4 ^a,b,c^		6.41	13.3 ^a,b^		7.53		8.5 ^a,b^		5.14
December	4.9 ^ab^		3.56		2.6 ^a^		2.54	4.3 ^a^		2.97		3.5 ^a^		2.65
**Total**	**20.5**		**28.31**		**19.1**		**28.76**	**41.5**		**59.51**		**53.5**		**85.16**
**Month**		**2013**				**2014**					**2015**			
		**Mean**		**s**		**Mean**			**s**		**Mean**		**s**	
January		2.6 ^a^		1.93		3.9 ^a^			2.97		2.8 ^a^		2.32	
February		2.6 ^a^		2.26		4.6 ^a,b^			3.42		2.7 ^a^		2.01	
March		4.9 ^a,b^		3.70		7.0 ^a,b^			5.13		7.3 ^a,b^		4.80	
April		10.3 ^a,b,c^		7.89		21.2 ^c^			11.13		14.9 ^c^		7.96	
May		13.5 ^c^		8.81		23.5 ^c^			10.94		30.0 ^d^		14.86	
June		29.0 ^d^		18.51		64.7 ^e^			32.46		81.7 ^g^		40.25	
July		92.9 ^g^		41.65		63.2 ^e^			31.65		68.0 ^f^		31.04	
August		64.1 ^f^		30.38		60.1 ^e^			34.56		52.3 ^e^		24.41	
September		45.9 ^e^		26.47		52.1 ^d^			20.89		35.5 ^d^		14.97	
October		34.1 ^d^		17.48		28.5 ^c^			10.57		15.6 ^c^		6.50	
November		11.1 ^b,c^		5.67		12.4 ^b^			7.26		10.8 ^b,c^		6.38	
December		3.0 ^a^		2.52		4.6 ^a,b^			2.85		5.1 ^a,b^		3.40	
**Total**		**24.7**		**32.26**		**27.6**			**29.17**		**26.4**		**31.17**	

Different superindexes indicate significative differences (*p* < 0.05) according the Duncan post hoc test. s is standard deviation.

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
