# Peer review of "Surveillance of a Pest Through a Public Health Information System: The Case of the Blackfly (Simulium erythrocephalum) in Zaragoza (Spain) during 2009–2015"

_ijerph, 2020, doi:10.3390/ijerph17103734_

Round 1

Reviewer 1 Report

Authors have presented a nice study for understanding how the use of medical data and analytical tools can enhance understanding of simuliid pest problems affecting public health. The study is well designed and the Discussion presents a nice synthesis of information.  My primary recommendation is that authors improve the clarity of their presentation to avoid confusion and problems with interpretation and understanding, according to the following 9 points:

1.  Please correct the English.  As the manuscript currently reads, the English makes the reading and understanding difficult.

2.  The manuscript has numerous acronyms that makes reading more difficult (PC, BHA, etc).  At least PC could be spelled out as primary care each time.

3.  Abstract:  For the Conclusions statement, please provide actual benefits that the study provides, rather than stating it is “very useful” and “interesting”.

4.  Use of commas and periods in numbers to indicate decimals is confusing.  Please convert commas to periods when presenting decimals.  For example in Table 1, p < 0,001 should be p < 0.001 and 20,49 should be 20.49.  In fact, more appropriately the means should be carried to only one decimal place; thus, 20.5 (not 20.49).  In the tables, what is “s”?  If it is standard deviation or standard error (or something else?), it can appropriately be carried to 2 decimal places (e.g., 28.31), but please tell readers what “s” stands for.  In the footnote of Table 1, “test to test” is awkward.  Please rephrase.  In the caption of Table 2, the number “10.000 PC” is given, but should probably be “10,000 PC”?

5.  Please state the geographic reference area for Figs. 2 and 3, for example, metropolitan area of Zaragoza in Fig. 2.

6.  Figure 4:  “Monthly evolution” should be “Monthly numbers”. Presumably the x-axis values represent year and month (200901 would be January 2009?)  If so, please state so.  The figure seems not to correspond with different generations of blackflies.  The figure shows only one peak each year separated by winter.  interesting.  Do authors have an explanation in the Discussion?  Also, in the Discussion it will be helpful to say something about number of generations of the pest species. Presumably there are multiple generations?

7.  In the Materials and Methods and/or Results, please provide more information about the identity of the pest(s) of Simuliidae as it relates to the specific study results.  Simulium erythrocephalum is mentioned in the Introduction and Discussion, but not in Materials and Methods or Results.  If no identifications were made of the species actually responsible for any of the bites in this study, please state this and indicate that the responsible species is inferred to be S. erythrocephalum based on…  If there is a possibility that some other species might be involved, even in small numbers, please also indicate this.  For example, subgenus Wilhelmia is mentioned in the Discussion but its role as a biting pest of humans is not discussed.

8.  In Discussion, it will be helpful if authors can propose a hypothesis as to why results vary spatially from year to year? For example, are pest problems affected by wind direction?

9. Conclusions.  As in Abstract, it would be more helpful to state the actual outcomes and benefits of the study rather than telling readers that the tool is “very useful” and that the study “improved our knowledge”.  Of course, all good studies are useful and improve knowledge.  Please provide a stronger and more compelling Conclusion, such as how the results of the study can guide public health responses.

Author Response

Thank you very much for your useful comments to improve our manuscript. Please find below the modifications (in red in the manuscript) that we have included in the text. 

  1. 1.  Please correct the English. As the manuscript currently reads, the English makes the reading and understanding difficult.

We have improved the manuscript, corrected the English grammatical errors detected.

  1. The manuscript has numerous acronyms that makes reading more difficult (PC, BHA, etc).  At least PC could be spelled out as primary care each time.

As requested, we have spelled out PC as primary care in the text.

  1. Abstract:  For the Conclusions statement, please provide actual benefits that the study provides, rather than stating it is “very useful” and “interesting”.

As requested, we have modified the conclusions in the abstract but we cannot extend more because of the maximum of 200 words. It is as follows: This tool was very useful to describe the initial stage of this public health problem. It could be used for guiding public health responses in terms of surveillance and control management of this pest.

  1. Use of commas and periods in numbers to indicate decimals is confusing.  Please convert commas to periods when presenting decimals.  For example in Table 1, p < 0,001 should be p < 0.001 and 20,49 should be 20.49. 

We have corrected it.

In fact, more appropriately the means should be carried to only one decimal place; thus, 20.5 (not 20.49). In the tables, what is “s”?  If it is standard deviation or standard error (or something else?), it can appropriately be carried to 2 decimal places (e.g., 28.31), but please tell readers what “s” stands for.

We have only included one decimal in the mean and two for the s (standard deviation) in both tables. We have also included the meaning of s in the footnote of both tables.

In the footnote of Table 1, “test to test” is awkward.  Please rephrase.  In the caption of Table 2, the number “10.000 PC” is given, but should probably be “10,000 PC”?

We have corrected it.

  1. Please state the geographic reference area for Figs. 2 and 3, for example, metropolitan area of Zaragoza in Fig. 2.

As requested, we have included “metropolitan area of Zaragoza in Figures 2 and 3.

  1. Figure 4: “Monthly evolution” should be “Monthly numbers”. Presumably the x-axis values represent year and month (200901 would be January 2009?)  If so, please state so.

As requested we have change “monthly numbers” instead “monthly evolution”. We have also modified the X-axis values in the Figure 4 as requested.

The figure seems not to correspond with different generations of blackflies.  The figure shows only one peak each year separated by winter.  interesting.  Do authors have an explanation in the Discussion?  Also, in the Discussion it will be helpful to say something about number of generations of the pest species. Presumably there are multiple generations?

We agree with the reviewer. The monthly evolution of the data only show one peak each year. But when we did the weekly evolution analysis we found different peaks along the year that suggests the existence of several generations per year. In fact, the main peak we saw in the monthly evolution are in fact four different peaks that we think maybe correspond with five different generations of S. erythrocephalum. The pattern is not the same among years but the peaks normally have a distance between 2-3 weeks from one to another. When we did the same analysis adjusted relative to data from years pre-outbreak (2009 and 2010) we also saw little peaks out the spring-summer season pointed out more generations. It is only a hypothesis but we are aware that probably in Spain S. erythrocephalum have more than five generations per year, number pointed out by different authors in Germany, Slovakia, UK, Serbia and Lithuania.

The main reason for not including these results is that the manuscript would be too long. In fact, this information will be showed in another manuscript (in preparation) along with the analysis by age, sex and date of attendance. So, in the last sentence of the discussion, we pointed out that more aspects should be explored and to take into account what the reviewer has suggested we have added the following information ”… and improve our knowledge of the dynamics of S. erythrocephalum such as number of generations per year, among others.”

  1. In the Materials and Methods and/or Results, please provide more information about the identity of the pest(s) of Simuliidae as it relates to the specific study results.  Simulium erythrocephalumis mentioned in the Introduction and Discussion, but not in Materials and Methods or Results.  If no identifications were made of the species actually responsible for any of the bites in this study, please state this and indicate that the responsible species is inferred to be S. erythrocephalum based on…  If there is a possibility that some other species might be involved, even in small numbers, please also indicate this.  For example, subgenus Wilhelmia is mentioned in the Discussion but its role as a biting pest of humans is not discussed.

As suggested, we have included some information related to S. erythrocephalum in the study area section in the Materials and methods. It is as follows: “No identifications of blackfly species were made in this study. Responsible species of human bites in Zaragoza is inferred to be S. erythrocephalum based on previous studies carried out during our same study period [7,17]. Species of subgenus Wilhelmia have been also detected in the outskirts of Zaragoza city but they have not been identified in human biting [7] so its role as biting pest of humans is not discussed.”

  1. In Discussion, it will be helpful if authors can propose a hypothesis as to why results vary spatially from year to year? For example, are pest problems affected by wind direction?

We tried to explain this request in lines 228-237: “A plausible explanation for the fact that urban areas have fewer insect bites records than rural zones during the 2013-2015 period is that the main breeding sites for S. erythrocephalum in the Ebro River are down- and upstream the urban area of Zaragoza [7]. A significant number of consultations in rural BHA far from the Ebro River but where the Huerva and Gállego Rivers flow suggests that these latter rivers have blackfly breeding sites and the insect bites medical records in these BHA are strongly influenced by these rivers and not by the Ebro. Larvae and pupae of S. erythrocephalum and other species belonging to the subgenus Wilhelmia has been identified in both rivers outside the urban area of Zaragoza city [23]. However, the BHA through which the urban section of the Huerva River runs are not as affected by insect bites (Figure 3) because the conditions for the development of the preimaginal forms do not exist [17]. “

The authors don’t believe that the wind direction is a factor affecting the overall spatial pattern of S. erythrocephalum females in the study area. The wind probably affect the recolonization of urban BHA from the main breeding sites in rural BHA in windy days but not generally. In BHA in the Ebro axis, a possible hypothesis could be that the production of females is lower in years where there are less breeding sites available and therefore the smaller number of emerged females may continue to bite in areas where they are hatched (mainly in rural BHA) and the displacement to urban areas is less because there is much less total population. Anyway, it is only a hypothesis and we did not want to hypothesize on this aspect without having more consistent results.

  1. Conclusions.  As in Abstract, it would be more helpful to state the actual outcomes and benefits of the study rather than telling readers that the tool is “very useful” and that the study “improved our knowledge”.  Of course, all good studies are useful and improve knowledge.  Please provide a stronger and more compelling Conclusion, such as how the results of the study can guide public health responses.

Regarding to the conclusions, we have improved it and the paragraph is as follows: “The systematic analysis of the electronic medical records of the OMI-AP system has shown to be a very useful tool to enable the early identification of a public health problem at the regional level through the epidemiology of arthropod bites in primary care. Furthermore, this study can be used to guide public health responses like to assess changes in arthropod biting and to establish highest risk blackfly biting periods. This study could establish the bases of a syndromic surveillance system to help taking blackfly control decisions, such as establishing a baseline incidence of arthropod bites that marks the beginning of the control season and evaluate the blackfly pest management.”

Reviewer 2 Report

The manuscript provides useful knowledge about the simuliid biting activity in in and around Zaragoza City, Spain, gleaned from a bite records from a regional medical information system. Because a black fly nuisance problem involving only a handful of species has been well established in the region with its large rivers, it is likely not too large of a stretch to implicate black flies, rather than other biting Diptera, as the cause. Overall, I think this approach and conclusions drawn from these data have validity and should appear in the primary literature. I have attached the manuscript, upon which I have corrected a good deal of English grammatical errors. I believe the Abstract should be revisited to make it completely standalone; it is not in its current form.

Author Response

Thank you very much for your useful comments to improve our manuscript. Please find below the modifications (in red in the manuscript) that we have included in the text.

  • Lines 14 and 40: At the European level is never explained in the text. I have never heard this jargon term. We meant that Zaragoza is currently the city in Europe more affected by blackfly pest in number of human bites. As ssugested we have corrected the sentence and included “in Europe” instead of “European level”.

Regarding the use of an unexplained acronym (OMI-AP) in an Abstract, we have included the company and the country after the first appearence of OMI-AP in the text because OMI-AP is a software implemented by the Government of Aragón.

  • Regarding to the Abstract that should be revisited to make it completely standalone and it is not in its current form.

As requested we have removed the numbers but we have followed the manuscript template given in the journal where the abstract is separated in different subheadings.

  • Sentence structure is poor in the abstract“…this tool highlighted very useful to describe the beginnning”…

As requested, we have modified the conclusions in the abstract but we cannot extend more because of the maximum of 200 words. It is as follows: “This tool was very useful to describe the initial stage of this public health problem. It could be used for guiding public health responses in terms of surveillance and management of this pest.”

  • Line 79 These terms are contradictory “Strong low waters”.

It has been corrected to “very low waters”.

  • Line 89. Incomplete sentence. We have modified the sentence as” Both data were disaggregated by BHA belonging to the metropolitan area of Zaragoza city in the period 2009-2015.”

We have also included all corrections related with English grammatical errors.